# NET-DNF: EFFECTIVE DEEP MODELING OF TABULAR DATA

**Liran Katzir**  
lirank@google.com

Gal Elidan  
elidan@google.com

Ran El-Yaniv  
rani@cs.technion.ac.il

## ABSTRACT

A challenging open question in deep learning is how to handle tabular data. Unlike domains such as image and natural language processing, where deep architectures prevail, there is still no widely accepted neural architecture that dominates tabular data. As a step toward bridging this gap, we present Net-DNF a novel generic architecture whose inductive bias elicits models whose structure corresponds to logical Boolean formulas in disjunctive normal form (DNF) over affine soft-threshold decision terms. Net-DNFs also promote localized decisions that are taken over small subsets of the features. We present extensive experiments showing that Net-DNFs significantly and consistently outperform fully connected networks over tabular data. With relatively few hyperparameters, Net-DNFs open the door to practical end-to-end handling of tabular data using neural networks. We present ablation studies, which justify the design choices of Net-DNF including the inductive bias elements, namely, Boolean formulation, locality, and feature selection.

## 1 INTRODUCTION

A key point in successfully applying deep neural models is the construction of architecture families that contain inductive bias relevant to the application domain. Architectures such as CNNs and RNNs have become the preeminent favorites for modeling images and sequential data, respectively. For example, the inductive bias of CNNs favors locality, as well as translation and scale invariances. With these properties, CNNs work extremely well on image data, and are capable of generating problem-dependent representations that almost completely overcome the need for expert knowledge. Similarly, the inductive bias promoted by RNNs and LSTMs (and more recent models such as transformers) favors both locality and temporal stationarity.

When considering *tabular data*, however, neural networks are not the hypothesis class of choice. Most often, the winning class in learning problems involving *tabular data* is decision forests. In Kaggle competitions, for example, gradient boosting of decision trees (GBDTs) (Chen & Guestrin, 2016; Friedman, 2001; Prokhorenkova et al., 2018; Ke et al., 2017) are generally the superior model. While it is quite practical to use GBDTs for medium size datasets, it is extremely hard to scale these methods to very large datasets. Scaling up the gradient boosting models was addressed by several papers (Ye et al., 2009; Tyree et al., 2011; Fu et al., 2019; Vasiloudis et al., 2019). The most significant computational disadvantage of GBDTs is the need to store (almost) the entire dataset in memory[1]. Moreover, handling multi-modal data, which involves both tabular and spatial data (e.g., medical records and images), is problematic. Thus, since GBDTs and neural networks cannot be organically optimized, such multi-modal tasks are left with sub-optimal solutions. The creation of a purely neural model for tabular data, which can be trained with SGD end-to-end, is therefore a prime open objective.

A few works have aimed at constructing neural models for tabular data (see Section 5). Currently, however, there is still no widely accepted end-to-end neural architecture that can handle tabular data and consistently replace fully-connected architectures, or better yet, replace GBDTs. Here we present Net-DNFs, a family of neural network architectures whose primary inductive bias is an ensemble comprising a disjunctive normal form (DNF) formulas over linear separators. This family also promotes (input) feature selection and spatial localization of ensemble members. These inductive

---

[1]This disadvantage is shared among popular GBDT implementations: XGBoost, LightGBM, and CatBoost.

biases have been included by design to promote conceptually similar elements that are inherent in GBDTs and random forests. Appealingly, the Net-DNF architecture can be trained end-to-end using standard gradient-based optimization. Importantly, it consistently and significantly outperforms FCNs on tabular data, and can sometime even outperform GBDTs.

The choice of appropriate inductive bias for specialized hypothesis classes for tabular data is challenging since, clearly, there are many different kinds of such data. Nevertheless, the "universality" of forest methods in handling a wide variety of tabular data suggests that it might be beneficial to emulate, using neural networks, the important elements that are part of the tree ensemble representation and algorithms. Concretely, every decision tree is equivalent to some DNF formula over axis-aligned linear separators (see details in Section 3). This makes DNFs an essential element in any such construction. Secondly, all contemporary forest ensemble methods rely heavily on feature selection. This feature selection is manifested both during the induction of each individual tree, where features are sequentially and greedily selected using information gain or other related heuristics, and by uniform sampling features for each ensemble member. Finally, forest methods include an important localization element – GBDTs with their sequential construction within a boosting approach, where each tree re-weights the instance domain differently – and random forests with their reliance on bootstrap sampling. Net-DNFs are designed to include precisely these three elements.

After introducing Net-DNF, we include a Vapnik-Chervonenkins (VC) comparative analysis of DNFs and trees showing that DNFs potentially have advantage over trees when the input dimension is large and vice versa. We then present an extensive empirical study. We begin with an ablation study over three real-life tabular data prediction tasks that convincingly demonstrates the importance of all three elements included in the Net-DNF design. Second, we analyze our novel feature selection component over controlled synthetic experiments, which indicate that this component is of independent interest. Finally, we compare Net-DNFs to FCNs and GBDTs over several large classification tasks, including two past Kaggle competitions. Our results indicate that Net-DNFs consistently outperform FCNs, and can sometime even outperform GBDTs.

## 2 DISJUNCTIVE NORMAL FORM NETWORKS (NET-DNFS)

In this section we introduce the Net-DNF architecture, which consists of three elements. The main component is a block of layers emulating a DNF formula. This block will be referred to as a *Disjunctive Normal Neural Form* (DNNF). The second and third components, respectively, are a feature selection module, and a localization one. In the remainder of this section we describe each component in detail. Throughout our description we denote by $\mathbf{x} \in \mathbb{R}^d$ a column of input feature vectors, by $\mathbf{x}_i$, its $i$th entry, and by $\sigma(\cdot)$ the sigmoid function.

### 2.1 A DISJUNCTIVE NORMAL NEURAL FORM (DNNF) BLOCK

A *disjunctive normal neural form* (DNNF) block is assembled using a two-hidden-layer network. The first layer creates affine "literals" (features) and is trainable. The second layer implements a number of soft conjunctions over the literals, and the third output layer is a neural OR gate. Importantly, only the first layer is trainable, while the two other are binary and fixed.

We begin by describing the neural AND and OR gates. For an input vector $\mathbf{x}$, we define soft, differentiable versions of such gates as

$$\text{OR}(\mathbf{x}) \triangleq \tanh\left(\sum_{i=1}^{d} \mathbf{x}_i + d - 1.5\right), \qquad \text{AND}(\mathbf{x}) \triangleq \tanh\left(\sum_{i=1}^{d} \mathbf{x}_i - d + 1.5\right).$$

These definitions are straightforwardly motivated by the precise neural implementation of the corresponding binary gates. Notice that by replacing $\tanh$ by a binary activation and changing the bias constant from 1.5 to 1, we obtain an exact implementation of the corresponding logical gates for binary input vectors (Anthony, 2005; Shalev-Shwartz & Ben-David, 2014); see a proof of this statement in Appendix A. Notably, each unit does not have any trainable parameters. We now define the AND gate in a vector form to project the logical operation over a subset of variables. The projection is controlled by an indicator column vector (a mask) $\mathbf{u} \in \{0, 1\}^d$. With respect to such a projection vector $\mathbf{u}$, we define the corresponding *projected* gate as $\text{AND}_{\mathbf{u}}(\mathbf{x}) \triangleq \tanh\left(\mathbf{u}^T\mathbf{x} - ||\mathbf{u}||_1 + 1.5\right)$.

Equipped with these definitions, a $\mathrm{DNNF}(\mathbf{x}) : \mathbb{R}^d \rightarrow \mathbb{R}$ with $k$ conjunctions over $m$ literals is,

$$L(\mathbf{x}) \triangleq \tanh(\mathbf{x}^T W + \mathbf{b}) \in \mathbb{R}^m \tag{1}$$

$$\mathrm{DNNF}(\mathbf{x}) \triangleq \mathrm{OR}([\mathrm{AND}_{\mathbf{c}^1}(L(\mathbf{x})), \mathrm{AND}_{\mathbf{c}^2}(L(\mathbf{x})), \ldots, \mathrm{AND}_{\mathbf{c}^k}(L(\mathbf{x}))]). \tag{2}$$

Equation (1) defines $L(\mathbf{x})$ that generates $m$ "neural literals", each of which is the result of a $\tanh$-activation of a (trainable) affine transformation. The (trainable) matrix $W \in \mathbb{R}^{d \times m}$, as well as the row vector bias term $\mathbf{b} \in \mathbb{R}^m$, determine the affine transformations for each literal such that each of its columns corresponds to one literal. Equation (2) defines a DNNF. In this equation, the vectors $\mathbf{c}^i \in \{0, 1\}^m$, $1 \leq i \leq k$, are binary indicators such that $\mathbf{c}_j^i = 1$ iff the $j$th literal belongs to the $i$th conjunction. In our design, each literal belongs to a single conjunction. These indicator vectors are defined and fixed according to the number and length of the conjunctions (See Appendix D.2).

## 2.2 NET-DNFs

The embedding layer of a Net-DNF with $n$ DNNF blocks is a simple concatenation

$$E(\mathbf{x}) \triangleq [\mathrm{DNNF}_1(\mathbf{x}), \mathrm{DNNF}_2(\mathbf{x}), \ldots, \mathrm{DNNF}_n(\mathbf{x})]. \tag{3}$$

Depending on the application, the final Net-DNF is a composition of an output layer over $E(\mathbf{x})$. For example, for binary classification (logistic output layer), $\mathrm{Net\text{-}DNF}(x) : \mathbb{R}^d \rightarrow (0, 1)$ is,

$$\mathrm{Net\text{-}DNF}(\mathbf{x}) \triangleq \sigma\left(\sum_{i=1}^{n} w_i \mathrm{DNNF}_i(\mathbf{x}) + b_i\right). \tag{4}$$

To summarize, a Net-DNF is always a four-layer network (including the output layer), and only the first and last layers are learned. Each DNNF block has two parameters: the number of conjunctions $k$ and the length $m$ of these conjunctions, allowing for a variety of Net-DNF architectures. In all our experiments we considered a single Net-DNF architecture that has a fixed diversity of DNNF blocks which includes a number of different DNNF groups with different $k$, each of which has a number of conjunction sizes $m$ (see details in Appendix D.2). The number $n$ of DNNFs was treated as a hyperparameter, and selected based on a validation set as described on Appendix D.1.

## 2.3 FEATURE SELECTION

One key strategy in decision tree training is greedy feature selection, which is performed hierarchically at any split, and allows decision trees to exclude irrelevant features. Additionally, decision tree ensemble algorithms apply random sampling to select a subset of the features, which is used to promote diversity, and prevent different trees focusing on the same set of dominant features in their greedy selection. In line with these strategies, we include in our Net-DNFs conceptually similar feature selection elements: (1) a subset of features uniformly and randomly sampled for each DNNF; (2) a trainable mechanism for feature selection, applied on the resulting random subset. These two elements are combined and implemented in the affine literal generation layer described in Equation (1), and applied independently for each DNNF. We now describe these techniques in detail.

Recalling that $d$ is the input dimension, the random selection is made by generating a stochastic binary mask, $\mathbf{m}_s \in \{0, 1\}^d$ (each block has its own mask), such that the probability of any entry being 1 is $p$ (see Appendix D.2 for details on setting this parameter). For a given mask $\mathbf{m}_s$, this selection can be applied over affine literals using a simple product $\mathrm{diag}(\mathbf{m}_s)W$, where $W$ is the matrix of Equation (1). We then construct a *trainable* mask $\mathbf{m}_t \in \mathbb{R}^d$, which will be applied on the features that are kept by $\mathbf{m}_s$. We introduce a novel trainable feature selection component that combines binary quantization of the mask together with modified elastic-net regularization. To train a binarized vector we resort to the straight-through estimator (Hinton, 2012; Hubara et al., 2017), which can be used effectively to train non-differentiable step functions such as a threshold or sign. The trick is to compute the step function *exactly* in the forward pass, and utilize a differentiable proxy in the backward pass. We use a version of the straight-through estimator for the sign function (Bengio et al., 2013),

$$\Phi(x) \triangleq \begin{cases} \mathrm{sign}(x), & \text{forward pass;} \\ \tanh(x), & \text{backward pass.} \end{cases}$$

Using the estimator $\Phi(x)$, we define a differentiable binary threshold function $T(x) = \frac{1}{2}\Phi(|x|-\epsilon)+\frac{1}{2}$, where $\epsilon \in \mathbb{R}$ defines an epsilon neighborhood around zero for which the output of $T(x)$ is zero, and one outside of this neighborhood (in all our experiments, we set $\epsilon = 1$ and initialize the entries of $\mathbf{m}_t$ above this threshold). We then apply this selection by $\operatorname{diag}(T(\mathbf{m}_t))W$. Given a fixed stochastic selection $\mathbf{m}_s$, to train the binarized selection $\mathbf{m}_t$ we employ regularization. Specifically, we consider a modified version of the elastic net regularization, $R(\mathbf{m}_t, \mathbf{m}_s)$, which is tailored to our task. The modifications are reflected in two parts. First, the balancing between the $L_1$ and $L_2$ regularization is controlled by a trainable parameter $\alpha \in \mathbb{R}$. Second, the expressions of the $L_1$ and $L_2$ regularization are replaced by $R_1(\mathbf{m}_t, \mathbf{m}_s), R_2(\mathbf{m}_t, \mathbf{m}_s)$, respectively (defined below). Moreover, since we want to take into account only features that were selected by the random component, the regularization is applied on the vector $\mathbf{m}_{ts} = \mathbf{m}_t \odot \mathbf{m}_s$, where $\odot$ is element-wise multiplication. The functional form of the modified elastic net regularization is as follows,

$$R_2(\mathbf{m}_t, \mathbf{m}_s) \triangleq \left| \frac{||\mathbf{m}_{ts}||_2^2}{||\mathbf{m}_s||_1} - \beta\epsilon^2 \right|, \qquad R_1(\mathbf{m}_t, \mathbf{m}_s) \triangleq \left| \frac{||\mathbf{m}_{ts}||_1}{||\mathbf{m}_s||_1} - \beta\epsilon \right|$$

$$R(\mathbf{m}_t, \mathbf{m}_s) \triangleq \frac{1-\sigma(\alpha)}{2}R_2(\mathbf{m}_t, \mathbf{m}_s) + \sigma(\alpha)R_1(\mathbf{m}_t, \mathbf{m}_s).$$

The above formulation of $R_2(\cdot)$ and $R_1(\cdot)$ is motivated as follows. First, we normalize both norms by dividing with the effective input dimension, $||\mathbf{m}_s||_1$, which is done to be invariant to the (effective) input size. Second, we define $R_2$ and $R_1$ as absolute errors, which encourages each entry to be, on average, approximately equal to the threshold $\epsilon$. The reason is that the vector $\mathbf{m}_t$ passes through a binary threshold, and though the exact values of its entries are irrelevant. What is relevant is whether these values are within epsilon neighborhood of zero or not. Thus, when the values are roughly equal to the threshold, it is more likely to converge to a balanced point where the regularization term is low and the relevant features were selected. The threshold term is controlled by $\beta$ (a hyperparameter), which controls the cardinality of $\mathbf{m}_t$, where smaller values of $\beta$ lead to sparser $\mathbf{m}_t$. To summarize, feature selection is manifested by both architecture and loss. Architecture relies on the masks $m_t, m_s$, while the loss function uses $R(m_t, m_s)$.

Finally, the functional form of a DNNF block with the feature selection component is obtained by plugging the masks into Equation (2), $L(\mathbf{x}) \triangleq \tanh(\mathbf{x}^T \operatorname{diag}(T(\mathbf{m}_t)) \operatorname{diag}(\mathbf{m}_s)W + \mathbf{b}) \in \mathbb{R}^m$. Additionally, the mean over $R(\mathbf{m}_t, \mathbf{m}_s)$ in all DNNFs is added to the loss function as a regularizer.

## 2.4 SPATIAL LOCALIZATION

The last element we incorporate in the Net-DNF construction is *spatial localization*. This element encourages each DNNF unit in a Net-DNF ensemble to specialize in some focused proximity of the input domain. Localization is a well-known technique in classical machine learning, with various implementations and applications (Jacobs et al., 1991; Meir et al., 2000). On the one hand, localization allows construction of low-bias experts. On the other hand, it helps promote diversity, and reduction of the correlation between experts, which can improve the performance of an ensemble (Jacobs, 1997; Derbeko et al., 2002). We incorporate spatial localization by associating a Gaussian kernel $\operatorname{loc}(\mathbf{x}|\mu, \mathbf{\Sigma})_i$ with a trainable mean vector $\mu_i$ and a trainable diagonal covariance matrix $\mathbf{\Sigma}_i$ for the $i$th DNNF. Given a Net-DNF with $n$ DNNF blocks, the functional form of its embedding layer (Equation 3), with the spatial localization, is

$$\operatorname{loc}(\mathbf{x}|\mu, \mathbf{\Sigma}) \triangleq [e^{-||\mathbf{\Sigma}_1(\mathbf{x}-\mu_1)||_2}, e^{-||\mathbf{\Sigma}_2(\mathbf{x}-\mu_2)||_2}, \ldots, e^{-||\mathbf{\Sigma}_n(\mathbf{x}-\mu_n)||_2}] \in \mathbb{R}^n$$

$$\operatorname{sm-loc}(\mathbf{x}|\mu, \mathbf{\Sigma}) \triangleq \operatorname{Softmax}\{\operatorname{loc}(\mathbf{x}|\mu, \mathbf{\Sigma}) \cdot \sigma(\tau)\} \in (0,1)^n$$

$$E(\mathbf{x}) \triangleq [\operatorname{sm-loc}(\mathbf{x}|\mu, \mathbf{\Sigma})_1 \cdot \operatorname{DNNF}_1(\mathbf{x}), \ldots, \operatorname{sm-loc}(\mathbf{x}|\mu, \mathbf{\Sigma})_n \cdot \operatorname{DNNF}_n(\mathbf{x})],$$

where $\tau \in \mathbb{R}$ is a trainable parameter such that $\sigma(\tau)$ serves as the trainable temperature in the softmax. The inclusion of an adaptive temperature in this localization mechanism facilitates a data-dependent degree of exclusivity: at high temperatures, only a few DNNFs will handle an input instance whereas at low temperatures, more DNNFs will effectively participate in the ensemble. Observe that our localization mechanism is fully trainable and does not add any hyperparameters.

## 3    DNFs and Trees – A VC Analysis

The basic unit in our construction is a (soft) DNF formula instead of a tree. Here we provide a theoretical perspective on this design choice. Specifically, we analyze the VC-dimension of Boolean DNF formulas and compare it to that of decision trees. With this analysis we gain some insight into the generalization ability of formulas and trees, and argue numerically that the generalization of a DNF can be superior to a tree when the input dimension is not small (and vice versa).

Throughout this discussion, we consider binary classification problems whose instances are Boolean vectors in $\{0, 1\}^n$. The first simple observation is that every decision tree has an equivalent DNF formula. Simply, each tree path from the root to a positively labeled leaf can be expressed by a conjunction of the conditions over the features appearing along the path to the leaf, and the whole tree can be represented by a disjunction of the resulting conjunctions. However, DNFs and decision trees are not equivalent, and we demonstrate that in the lense of VC-dimension. Simon (1990) presented an exact expression for the VC-dimension of decision trees as a function of the tree *rank*.

**Definition 1** (Rank). *Consider a binary tree $T$. If $T$ consists of a single node, its rank is defined as 0. If $T$ consists of a root, a left subtree $T_0$ of rank $r_0$, and a right subtree $T_1$ of rank $r_1$, then*

$$rank(T) = \begin{cases} 1 + r_0 & \text{if } r_0 = r_1 \\ \max\{r_0, r_1\} & else \end{cases}$$

Clearly, for any decision tree $T$ over $n$ variables, $1 \leq rank(T) \leq n$. Also, it is not hard to see that a binary tree $T$ has a rank greater than $r$ iff the complete binary tree of depth $r + 1$ can be embedded into $T$.

**Theorem 1** (Simon (1990)). *Let $DT_n^r$ denote the class of decision trees of rank at most $r$ on $n$ Boolean variables. Then it holds that $VCDim(DT_n^r) = \sum_{i=0}^r \binom{n}{i}$.*

The following theorem, whose proof appears in Appendix B, upper bounds the VC-dimension of a Boolean DNF formula.

**Theorem 2** (DNF VC-dimension bound). *Let $DNF_n^k$ be the class of DNF formulas with $k$ conjunctions on $n$ Boolean variables. Then it holds that $VCDim(DNF_n^k) \leq 2(n + 1)k \log(3k)$.*

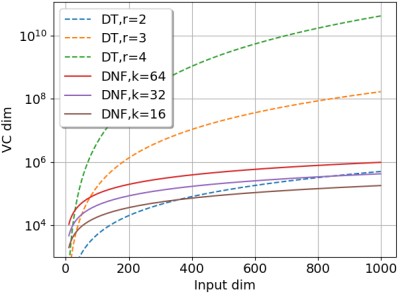

Figure 1: $VCDim(DT_n^r)$ and the upper bound on $VCDim(DNF_n^k)$ (log scale) as a function of the input dimension

It is evident that in the case of DNF formulas the upper bound on the VC-dimension grows linearly with the input dimension, whereas in the case of decision trees, if the rank is greater than 1, the VC-dimension grows polynomially (with degree at least 2) with the input dimension. In the worst case, this growth is exponential. A direct comparison of these dimensions is not trivial because there is a complex dependency between the rank $r$ of a decision tree, and the number $k$ of the conjunctions of an equivalent DNF formula. Even if we compare large-$k$ DNF formulas to small-rank trees, it is clear that the VC-dimension of the trees can be significantly larger. For example, in Figure 1, we plot the upper bounds on the VC-dimension of large formulas (solid curves), and the exact VC-dimensions of small-rank trees (dashed curves). With the exception of rank-2 trees, the VC-dimension of decision trees dominates the dimension of DNFs, when the input dimension exceeds 100. Trees, however, may have an advantage over DNF formulas for low-dimensional inputs. Since the VC-dimension is a qualitative proxy of the sample complexity of a hypothesis class, the above analysis provides theoretical motivation for expressing trees using DNF formulas when the input dimension is not small. Having said that, the disclaimer is that in the present discussion we have only considered binary problems. Moreover, the final hypothesis classes of both Net-DNFs and GBDTs are more complex in structure.

## 4    Empirical Study

In this section, we present an empirical study that substantiates the design of Net-DNFs and convincingly shows its significant advantage over FCN architectures. The datasets used in this study

are from Kaggle competitions and OpenML (Vanschoren et al., 2014). A summary of these datasets appears in Appendix C. All results presented in this work were obtained using a massive grid search for optimizing each model's hyperparameters. A detailed description of the grid search process with additional details can be found in Appendices D.1, D.2. We present the scores for each dataset according to the score function defined in the Kaggle competition we used, log-loss and area under ROC curve (AUC ROC) for multiclass datasets and binary datasets, respectively. All results are the mean of the test scores over five different partitions, and the standard error of the mean is reported.[2]

In addition, we also conducted a preliminary study of TabNet (Arik & Pfister, 2019) (see Section 5) over our datasets using its PyTorch implementation[3], but failed to produce competitive results.[4]

**The merit of the different Net-DNF components.** We start with two different ablation studies, where we evaluate the contributions of the three Net-DNF components. In the first study, we start with a vanilla three-hidden-layer FCN and gradually add each component separately. In the second study, we start each experiment with the complete Net-DNF and leave one component out each time. In each study, we present the results on three real-world datasets, where all results are test log-loss scores (lower is better), out-of-memory (OOM) entries mean that the network was too large to execute on our machine (see Appendix D.2). More technical details can be found in Appendix D.4.

Table 1: **Gradual study** (test log-loss scores)

| Dataset | Eye Movements | | | Gesture Phase | | | Gas Concentrations | | |
| # formulas | 128 | 512 | 2048 | 128 | 512 | 2048 | 128 | 512 | 2048 |
|---|---|---|---|---|---|---|---|---|---|
| Exp 1: Fully trained FCN | 0.9864 (±0.0038) | 1.0138 (±0.0083) | OOM | 1.3139 (±0.0067) | 1.3368 (±0.0084) | OOM | 0.2423 (±0.0598) | 0.3862 (±0.0594) | OOM |
| Exp 2: Adding DNF structure | 0.9034 (±0.0058) | 0.9336 (±0.0058) | 1.3011 (±0.0431) | 1.1391 (±0.0059) | 1.1812 (±0.0117) | 1.8633 (±0.1026) | 0.0351 (±0.0048) | 0.0421 (±0.0046) | 0.0778 (±0.0080) |
| Exp 3: Adding feature selection | 0.8134 (±0.0142) | 0.8163 (±0.0096) | 0.9652 (±0.0143) | 1.1411 (±0.0093) | 1.1320 (±0.0083) | 1.3015 (±0.0317) | 0.0227 (±0.0019) | 0.0265 (±0.0012) | 0.0516 (±0.0061) |
| Exp 4: Adding localization | 0.7621 (±0.0079) | 0.7125 (±0.0077) | **0.6903** (±0.0049) | 0.9742 (±0.0079) | 0.9120 (±0.0123) | **0.8770** (±0.0088) | 0.0162 (±0.0013) | 0.0149 (±0.0008) | **0.0145** (±0.0011) |

Consider Table 1. In **Exp 1** we start with a vanilla three-hidden-layer FCN with a tanh activation. To make a fair comparison, we defined the widths of the layers according to the widths in the Net-DNF with the corresponding formulas. In **Exp 2**, we added the DNF structure to the networks from Exp 1 (see Section 2.1). In **Exp 3** we added the feature selection component (Section 2.3). It is evident that performance is monotonically improving, where the best results are clearly obtained on the complete Net-DNF (Exp 4). A subtle but important observation is that in all of the first three experiments, for all datasets, the trend is that the lower the number of formulas, the better the score. This trend is reversed in **Exp 4**, where the localization component (Section 2.4) is added, highlighting the importance of using all components of the Net-DNF representation in concert.

Now consider Table 2. In **Exp 5** we took the complete Net-DNF (Exp 4) and removed the feature selection component. When considering the Gesture Phase dataset, an interesting phenomenon is observed. In Exp 3 (128 formulas), we can see that the contribution of the feature selection component is negligible, but in Exp 5 (2048 formulas) we see the significant contribution of this component. We believe that the reason for this difference lies in the relationship of the feature selection component with the localization component, where this connection intensifies the contribution of the feature selection component. In **Exp 6** we took the complete Net-DNF (Exp 4) and removed the localization component (identical to Exp 3). We did the same in **Exp 7** where we removed the DNF structure. In general, it can be seen that removing each component results in a decrease in performance.

**An analysis of the feature selection component.** Having studied the contribution of the three components to Net-DNF, we now focus on the learnable part of the feature selection component (Section 2.3) alone, and examine its effectiveness using a series of synthetic tasks with a varying percentage of irrelevant features. Recall that when considering a single DNNF block, the feature

---

[2]Our code is available at https://github.com/amramabutbul/DisjunctiveNormalFormNet.

[3]https://github.com/dreamquark-ai/tabnet

[4]For example, for the Gas Concentration dataset (see below), TabNet results were slightly inferior to the results we obtained for XGBoost (4.89 log-loss for TabNet vs. 2.22 log-loss for XGBoost.

Table 2: **Leave one out study** (test log-loss scores)

| Dataset | Eye Movements | | | Gesture Phase | | | Gas Concentrations | | |
|---|---|---|---|---|---|---|---|---|---|
| # formulas | 128 | 512 | 2048 | 128 | 512 | 2048 | 128 | 512 | 2048 |
| Exp 4: Complete Net-DNF | 0.7621 (±0.0079) | 0.7125 (±0.0077) | **0.6903** (±0.0049) | 0.9742 (±0.0079) | 0.9120 (±0.0123) | **0.8770** (±0.0088) | 0.0162 (±0.0013) | 0.0149 (±0.0008) | **0.0145** (±0.0011) |
| Exp 5: Leave feature selection out | 0.8150 (±0.0046) | 0.8031 (±0.0046) | 0.7969 (±0.0054) | 0.9732 (±0.0082) | 0.9479 (±0.0081) | 0.9438 (±0.0111) | 0.0222 (±0.0018) | 0.0205 (±0.0021) | 0.0200 (±0.0022) |
| Exp 6: Leave localization out | 0.8134 (±0.0142) | 0.8163 (±0.0096) | 0.9652 (±0.0143) | 1.1411 (±0.0093) | 1.1320 (±0.0083) | 1.3015 (±0.0317) | 0.0227 (±0.0019) | 0.0265 (±0.0012) | 0.0516 (±0.0061) |
| Exp 7: Leave DNF structure out | 0.8403 (±0.0068) | 0.8128 (±0.0077) | OOM | 1.1265 (±0.0066) | 1.1101 (±0.0077) | OOM | 0.0488 (±0.0038) | 0.0445 (±0.0024) | OOM |

selection is a learnable binary mask that multiplies the input element-wise. Here we examine the effect of this mask on a vanilla FCN network (see technical details in Appendix D.5). The synthetic tasks we use were introduced by Yoon et al. (2019); Chen et al. (2018), where they were used as synthetic experiments to test feature selection. There are six different dataset settings; exact specifications appear in Appendix D.5. For each dataset, we generated seven different instances that differ in their input size. While increasing the input dimension $d$, the same logit is used for prediction, so the new features are irrelevant, and as $d$ gets larger, the percentage of relevant features becomes smaller.

We compare the performance of a vanilla FCN on three different cases: (1) oracle (ideal) feature selection (2) our (learned) feature selection mask, and (3) no feature selection. (See details in Appendix D.5). Consider the graphs in Figure 2, which demonstrate several interesting insights. In all tasks the performance of the vanilla FCN is sensitive to irrelevant features, probably due to the representation power of the FCN, which is prone to overfitting. On the other hand, by adding the feature selection component, we obtain near oracle performance on the first three tasks, and a significant improvement on the three others. Moreover, these results support our observation from the ablation studies: that the application of localization together with feature selection increases the latter's contribution. We can see that in Syn1-3 where there is a single interaction, the results are better than in Syn4-6 where the input space is divided into two 'local' sub-spaces with different interactions. These experiments emphasize the importance of the learnable feature selection in itself.

| Dataset | Test Metric | Net-DNF | XGBoost | FCN |
|---|---|---|---|---|
| Otto Group | log-loss | **45.600 ± 0.445** | 45.705 ± 0.361 | 47.898 ± 0.480 |
| Gesture Phase | log-loss | 86.798 ± 0.810 | **81.408 ± 0.806** | 102.070 ± 0.964 |
| Gas Concentrations | log-loss | **1.425 ± 0.104** | 2.219 ± 0.219 | 5.814 ± 1.079 |
| Eye Movements | log-loss | 68.037 ± 0.651 | **57.447 ± 0.664** | 78.797 ± 0.674 |
| Santander Transaction | roc auc | 88.668 ± 0.128 | **89.682 ± 0.165** | 86.722 ± 0.158 |
| House | roc auc | 95.451 ± 0.092 | **95.525 ± 0.138** | 95.164 ± 0.103 |

Table 3: Mean test results on tabular datasets and standard error of the mean. We present the ROC AUC (higher is better) as a percentage, and the log-loss (lower is better) with an x100 factor.

**Comparative Evaluation.** Finally, we compare the performance of Net-DNF vs. the baselines. Consider Table 3 where we examine the performance of Net-DNFs on six real-life tabular datasets (We add three larger datasets to those we used in the ablation studies). We compare our performance to XGboost Chen & Guestrin (2016), the widely used implementation of GBDTs, and to FCNs. For each model, we optimized its critical hyperparameters. This optimization process required many computational resources: thousands of configurations have been tested for FCNs, hundreds of configurations for XGBoost, and only a few dozen for Net-DNF. A detailed description of the grid search we used for each model can be found in Appendix D.3. In Table 3, we see that Net-DNF consistently and significantly outperforms FCN over all the six datasets. While obtaining better than or indistinguishable results from XGBoost over two datasets, on the other datasets, Net-DNF is slightly inferior but in the same ball park as XGBoost.

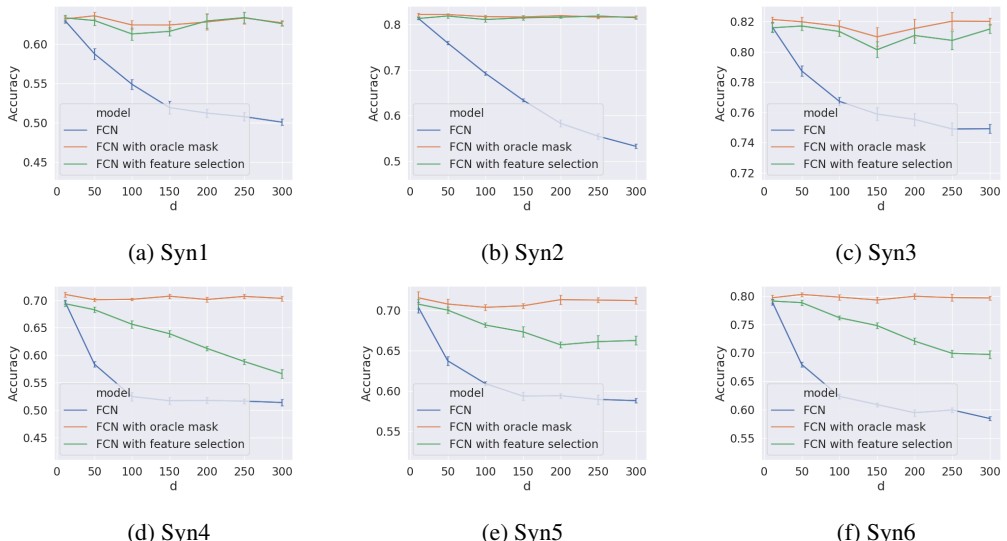

Figure 2: The results on the six synthetic experiments. For each experiment we present the test accuracy (with an error bar of the standard error of the mean) as a function of the input dimension $d$.

## 5 RELATED WORK

There have been a few attempts to construct neural networks with improved performance on tabular data. A recurring idea in some of these works is the explicit use of conventional decision tree induction algorithms, such as ID3 (Quinlan, 1979), or conventional forest methods, such as GBDT (Friedman, 2001) that are trained over the data at hand, and then parameters of the resulting decision trees are explicitly or implicitly "imported" into a neural network using teacher-student distillation (Ke et al., 2018), explicit embedding of tree paths in a specialized network architecture with some kind of DNF structure (Seyedhosseini & Tasdizen, 2015), and explicit utilization of forests as the main building block of layers (Feng et al., 2018). This reliance on conventional decision tree or forest methods as an integral part of the proposed solution prevents end-to-end neural optimization, as we propose here. This deficiency is not only a theoretical nuisance but also makes it hard to use such models on very large datasets and in combination with other neural modules.

A few other recent techniques aimed to cope with tabular data using pure neural optimization as we propose here. Yang et al. (2018) considered a method to approximate a single node of a decision tree using a soft binning function that transforms continuous features into one-hot features. While this method obtained results comparable to a single decision tree and an FCN (with two hidden layers), it is limited to settings where the number of features is small. Popov et al. (2019) proposed a network that combines elements of oblivious decision forests with dense residual networks. While this method achieved better results than GBDTs on several datasets, also FCNs achieved better than or indistinguishable results from GBDTs on most of these cases as well. Arik & Pfister (2019) presented TabNet, a neural architecture for tabular data that implements feature selection via sequential attention that offers instance-wise feature selection. It is reported that TabNet achieved results that are comparative or superior to GBDTs. Both TabNet and Net-DNF rely on sparsity inducing and feature selection, which are implemented in different ways. While TabNet uses an attention mechanism to achieve feature selection, Net-DNF uses DNF formulas and elastic net regularization. Focusing on microbiome data, a recent study Shavitt & Segal (2018) presented an elegant regularization technique, which produces extremely sparse networks that are suitable for microbiome tabular datasets. Finally, soft masks for feature selection have been considered before and the advantage of using elastic net regularization in a variable selection task was presented by Zou & Hastie (2005); Li et al. (2016).

## 6 CONCLUSIONS

We introduced Net-DNF, a novel neural architecture whose inductive bias revolves around a disjunctive normal neural form, localization and feature selection. The importance of each of these elements has been demonstrated over real tabular data. The results of the empirical study convincingly indicate that Net-DNFs consistently outperform FCNs over tabular data. While Net-DNFs do not consistently beat XGBoost, our results indicate that their performance score is not far behind GBDTs. Thus, Net-DNF offers a meaningful step toward effective usability of processing tabular data with neural networks

We have left a number of potential incremental improvements and bigger challenges to future work. First, in our work we only considered classification problems. We expect Net-DNFs to also be effective in regression problems, and it would also be interesting to consider applications in reinforcement learning over finite discrete spaces. It would be very interesting to consider deeper Net-DNF architectures. For example, instead of a single DNNF block, one can construct a stack of such blocks to allow for more involved feature generation. Another interesting direction would be to consider training Net-DNFs using a gradient boosting procedure similar to that used in XGBoost. Finally, a most interesting challenge that remains open is what would constitute the ultimate inductive bias for tabular prediction tasks, which can elicit the best architectural designs for these data. Our successful application of DNNFs indicates that soft DNF formulas are quite effective, and are strictly significantly superior to fully connected networks, but we anticipate that further effective biases will be identified, at least for some families of tabular tasks.

## ACKNOWLEDGMENTS

This research was partially supported by the Israel Science Foundation, grant No. 710/18.

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

## A   OR AND AND GATES

The (soft) neural OR and AND gates were defined as

$$
\mathrm{OR}(\mathbf{x}) \triangleq \tanh\left(\sum_{i=1}^{d} \mathbf{x}_i + d - 1.5\right), \qquad \mathrm{AND}(\mathbf{x}) \triangleq \tanh\left(\sum_{i=1}^{d} \mathbf{x}_i - d + 1.5\right).
$$

By replacing the $\tanh$ activation with a $\mathrm{sign}$ activation, and setting the bias term to 1 (instead of 1.5), we obtain exact binary gates,

$$
\mathrm{OR}(\mathbf{x}) \triangleq \mathrm{sign}\left(\sum_{i=1}^{d} \mathbf{x}_i + d - 1\right), \qquad \mathrm{AND}(\mathbf{x}) \triangleq \mathrm{sign}\left(\sum_{i=1}^{d} \mathbf{x}_i - d + 1\right).
$$

Consider a binary vector $\mathbf{x} \in \{\pm 1\}^d$. We prove that

$$
\mathrm{AND}(\mathbf{x}) \equiv \bigwedge_{i=1}^{d} \mathbf{x}_i,
$$

where, in the definition of the logical "and", $-1$ is equivalent to 0. If for any $1 \le i \le d$, $\mathbf{x}_i = 1$, then $\wedge_{i=1}^{d} \mathbf{x}_i = 1$. Conversely, we have,

$$
\mathrm{AND}(\mathbf{x}) = \sum_{i=1}^{d} \mathbf{x}_i - d + 1 = d - d + 1 = 1,
$$

and the application of the $\mathrm{sign}$ activation yields 1. In the case of the soft neural AND gate, we get $tanh(1) \approx 0.76$; therefore, we set the bias term to 1.5 to get an output closer to 1 ($tanh(1.5) \approx 0.9$).

Otherwise, there exists at least one index $1 \le j \le d$, such that $\mathbf{x}_j = -1$, and $\wedge_{i=1}^{d} \mathbf{x}_i = -1$. In this case,

$$
\mathrm{AND}(\mathbf{x}) = \sum_{i=1}^{d} \mathbf{x}_i - d + 1 = \mathbf{x}_j + \sum_{i \neq j} \mathbf{x}_i - d + 1 \le -1 + (d-1) - d + 1 = -1,
$$

and by applying the $\mathrm{sign}$ activation we obtain $-1$. This proves that the $\mathrm{AND}(\mathbf{x})$ neuron is equivalent to a logical "AND" gate in the binary case. A very similar proof shows that

$$
\mathrm{OR}(\mathbf{x}) \equiv \bigvee_{i=1}^{d} \mathbf{x}_i.
$$

## B   PROOF OF THEOREM 2

We bound the VC-dimension of a DNF formula in two steps. First, we derive an upper bound on the VC-dimension of a single conjunction, and then extend it to a disjunction of $k$ conjunctions. We use the following simple lemma.

**Lemma 1.** *For every two hypothesis classes, $H' \subseteq H$, it holds that $VCDim(H') \le VCDim(H)$.*

*Proof.* Let $d = VCDim(H')$. By definition, there exist $d$ points that can be shattered by $H'$. Therefore, there exist $2^d$ hypotheses $\{h_i'\}_{i=1}^{2^d}$ in $H'$, which shatter these points. By assumption, $\{h_i'\}_{i=1}^{2^d} \subseteq H$, so $VCDim(H) \ge d$. $\square$

For any conjunction on $n$ Boolean variables (regardless of the number of literals), it is possible to construct an equivalent decision tree of rank 1. The construction is straightforward. If $\bigwedge_{i=1}^{\ell} x_i$ is the conjunction, the decision tree consists of a single main branch of $\ell$ internal decision nodes connected sequentially. Each left child in this tree corresponds to decision "1", and each right child corresponds to decision "0". The root is indexed 1 and contains the literal $x_1$. For $1 \le i < \ell$, internal node $i$ contains the decision literal $x_i$ and its left child is node $i+1$ (whose decision literal is $x_{i+1}$). See the example in Figure 3.

It follows that the hypothesis class of conjunctions is contained in the class of rank-1 decision trees. Therefore, by Lemma 1 and Theorem 1, the VC-dimension of conjunctions is bounded above by $n + 1$.

We now derive the upper bound on the VC-dimension of a disjunction of $k$ conjunctions. Let $C$ be the class of conjunctions, and let $D_k(C)$ be the class of a disjunction of $k$ conjunctions. Clearly, $D_k(C)$ is a $k$-fold union of the class $C$, namely,

$$D_k(C) = \left\{ \bigcup_{i=0}^{k} c_i \mid c_i \in C \right\}.$$

By Lemma 3.2.3 in (Blummer et al. 1989), if $d = VCDim(C)$, then for all $k \geq 1$, $VCDim(D_k(C)) \leq 2dk \log(3k)$. Therefore, for the class $DNF_n^k$, of DNF formulas with $k$ conjunctions on $n$ Boolean variables, we have

$$VCDim(DNF_n^k) \leq 2(n+1)k \log(3k).$$

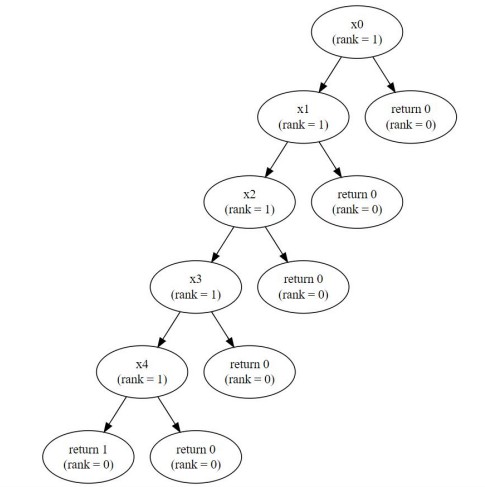

Figure 3: An example of a decision tree with rank 1, which is equivalent to the conjunction $x_0 \wedge x_1 \wedge x_2 \wedge x_3 \wedge x_4$.

## C    TABULAR DATASET DESCRIPTION

We use datasets (See Table 4) that differ in several aspects such as in the number of features (from 16 up to 200), the number of classes (from 2 up to 9), and the number of samples (from 10k up to 200k). To keep things simple, we selected datasets with no missing values, and that do not require preprocessing. All models were trained on the raw data without any feature or data engineering and without any kind of data balancing or weighting. Only feature wise standardization was applied.

| Dataset | features | classes | samples | source | link |
|---|---|---|---|---|---|
| Otto Group | 93 | 9 | 61.9k | Kaggle | kaggle.com/c/otto-group-product-classification-challenge/overview |
| Gesture Phase | 32 | 5 | 9.8k | OpenML | openml.org/d/4538 |
| Gas Concentrations | 129 | 6 | 13.9k | OpenML | openml.org/d/1477 |
| Eye Movements | 26 | 3 | 10.9k | OpenML | openml.org/d/1044 |
| Santander Transaction | 200 | 2 | 200k | Kaggle | kaggle.com/c/santander-customer-transaction-prediction/overview |
| House | 16 | 2 | 22.7k | OpenML | openml.org/d/821 |

Table 4: A description of the tabular datasets

## D    EXPERIMENTAL PROTOCOL

### D.1    DATA PARTITION AND GRID SEARCH PROCEDURE

All experiments in our work, using both synthetic and real datasets, were done through a grid search process. Each dataset was first randomly divided into five folds in a way that preserved the original distribution. Then, based on these five folds, we created five partitions of the dataset as follows. Each fold is used as the test set in one of the partitions, while the other folds are used as the training and validation sets. This way, each partition was 20% test, 10% validation, and 70% training. This division was done once [5], and the same partitions were used for all models. Based on these partitions, the following grid search process was repeated three times with three different seeds[6] (with the exact same five partitions as described before).

---

**Algorithm 1:** Grid Search Procedure

---

**Input:** model, configurations_list
results_list = [ ]
**for** *i=1 to n_partitions* **do**
    val_scores_list = [ ]
    test_scores_list = [ ]
    train, val, test = read_data(partition_index=i)
    **for** *c in configurations_list* **do**
        trained_model = model.train(train_data=train, val_data=val, configuration=c)
        trained_model.load_weights_from_best_epoch()
        val_score = trained_model.predict(data=val)
        test_score = trained_model.predict(data=test)
        val_scores_list.append(val_score)
        test_scores_list.append(test_score)
    **end**
    best_val_index = get_index_of_best_val_score(val_scores_list)
    test_res = test_scores_list[best_val_index]
    results_list.append(test_res)
**end**
mean = mean(results_list)
sem = standard_error_of_the_mean(results_list)
**Return:** mean, sem

---

The final mean and sem[7] that we presents in all experiments are the average across the three seeds. Additionally, as can be seen from Algorithm 1, the model that was trained on the training set (70%) is the one that is used to evaluate performance on the test set (20%). This was done to keep things simple. The loading wights command is relevant for the neural network models. While for the XGBoost, the framework handles the optimal number of estimators on prediction time (accordingly to early stopping on training time).

### D.2    TRAINING PROTOCOL

The Net-DNF and the FCN were implemented using Tesnorflow. To make a fair comparison, for both models, we used the same batch size[8] of 2048, and the same learning rate scheduler (reduce on plateau) that monitors the training loss. We set a maximum of 1000 epochs and used the same early stopping protocol (30 epochs) that monitors the validation score. Moreover, for both of them, we used the same loss function (softmax-cross-entropy for multi-class datasets and sigmoid-cross-entropy for binary datasets) and the same optimizer (Adam with default parameters).

---

[5]We used seed number 1.
[6]We used seed numbers 1, 2, 3.
[7]For details, see: docs.scipy.org/doc/scipy/reference/generated/scipy.stats.sem.html
[8]For Net-DNF , when using 3072 formulas, we set the batch size to 1024 on the Santander Transaction and Gas datasets and when using 2048 formulas, we set the batch size to 1024 on the Santander Transaction dataset. This was done due to memory issues.

For Net-DNF we used an initial learning rate of $0.05$. For FCN, we added the initial learning rate to the grid search with values of $\{0.05, 0.005, 0.0005\}$.

For XGBoost we set the maximal number of estimators to be 2500, and used an early stopping of 50 estimators that monitors the validation score.

All models were trained on GPUs - Titan Xp 12GB RAM.

Additionally, in the case of Net-DNF, we took a symmetry-breaking approach between the different DNNFs. This is reflected by the DNNF group being divided equally into four subgroups where, for each subgroup, the number of conjunctions is equal to one of the following values $[6, 9, 12, 15]$, and the group of conjunctions of each DNNF was divided equally into three subgroups where, for each subgroup, the conjunction length is equal to one of the following values $[2, 4, 6]$. The same approach was used for the parameter $p$ of the random mask. The DNNF group was divided equally into five subgroups where, for each subgroup, $p$ is equal to one of the following values $[0.1, 0.3, 0.5, 0.7, 0.9]$. In all experiments we used the same values.

### D.3 GRID PARAMETERS – TABULAR DATASETS

#### D.3.1 NET-DNF

| Net-DNF (42 configs) | |
| --- | --- |
| **hyperparameter** | **values** |
| number of formulas | $\{64, 128, 256, 512, 1024, 2048, 3072\}$ |
| feature selection beta | $\{1.6, 1.3, 1., 0.7, 0.4, 0.1\}$ |

#### D.3.2 XGBOOST

| XGBoost (864 configs) | |
| --- | --- |
| **hyperparameter** | **values** |
| number of estimators | $\{2500\}$ |
| learning rate | $\{0.001, 0.005, 0.01, 0.05, 0.1, 0.5\}$ |
| max depth | $\{2, 3, 4, 5, 7, 9, 11, 13, 15\}$ |
| colsample by tree | $\{0.25, 0.5, 0.75, 1.\}$ |
| sub sample | $\{0.25, 0.5, 0.75, 1.\}$ |

To summarize, we performed a crude but broad selection (among 42 hyper-parameter configurations) for our Net-DNF. Results were quite strong, so we avoided further fine tuning. To ensure extra fairness w.r.t. the baselines, we provided them with significantly more hyper-parameter tuning resources (864 configurations for XGBoost, and 3300 configurations for FCNs).

#### D.3.3 FULLY CONNECTED NETWORKS

The FCN networks are constructed using Dense-RELU-Dropout blocks with $L_2$ regularization. The network's blocks are defined in the following way. Given depth and width parameters, we examine two different configurations: (1) the same width is used for the entire network (e.g., if the width is 512 and the depth is four, then the network blocks are [512, 512, 512, 512]), and (2) the width parameter defines the width of the first block, and the subsequent blocks are reduced by a factor of 2 (e.g., if the width is 512 and the depth is four, then the network blocks are [512, 256, 128, 64]). On top of the last block we add a simple linear layer that reduce the dimension into the output dimension. The dropout and $L_2$ values are the same for all blocks.

| FCN (3300 configs) | |
| --- | --- |
| **hyperparameter** | **values** |
| depth | $\{1, 2, 3, 4, 5, 6\}$ |
| width | $\{128, 256, 512, 1024, 2048\}$ |
| $L_2$ lambda | $\{10^{-2}, 10^{-4}, 10^{-6}, 10^{-8}, 0.\}$ |
| dropout | $\{0., 0.25, 0.5, 0.75\}$ |
| initial learning rate | $\{0.05, 0.005, 0.0005\}$ |

## D.4 ABLATION STUDY

All ablation studies experiments were conducted using the grid search process as described in D.1. In all experiments, we used the same training details as described on D.2 for Net-DNF. Where the only difference between the different experiments is the addition or removal of the components.

The single hyperparameter that was fine-tuned using the grid search is the 'feature selection beta' on the range $\{1.6, 1.3, 1., 0.7, 0.4, 0.1\}$, in experiments in which the feature selection component is involved. In the other cases, only one configuration was tested in the grid search process for a specific number of formulas.

## D.5 FEATURE SELECTION ANALYSIS

The input features $\mathbf{x} \in \mathbb{R}^d$ of all six datasets were generated from a $d$-dimensional Gaussian distribution with no correlation across the features, $\mathbf{x} \sim \mathbb{N}(0, I)$. The label $\mathbf{y}$ is sampled as a Bernoulli random variable with $\mathbb{P}(\mathbf{y} = 1|\mathbf{x}) = \frac{1}{1+logit(\mathbf{x})}$, where $logit(\mathbf{x})$ is varied to create the different synthetic datasets ($\mathbf{x}_i$ refers to the $i$th entry):

1. **Syn1**: $logit(\mathbf{x}) = exp(\mathbf{x}_1 \mathbf{x}_2)$
2. **Syn2**: $logit(\mathbf{x}) = exp(\sum_{i=3}^{6} \mathbf{x}_i^2 - 4)$
3. **Syn3**: $logit(\mathbf{x}) = -10 \sin(2\mathbf{x}_7) + 2|\mathbf{x}_8| + \mathbf{x}_9 + exp(-\mathbf{x}_{10}) - 2.4$
4. **Syn4**: if $\mathbf{x}_{11} < 0$, logit follows **Syn1**, else, logit follows **Syn2**
5. **Syn5**: if $\mathbf{x}_{11} < 0$, logit follows **Syn1**, else, logit follows **Syn3**
6. **Syn6**: if $\mathbf{x}_{11} < 0$, logit follows **Syn2**, else, logit follows **Syn3**

We compare the performance of a basic FCN on three different cases: (1) **oracle (ideal) feature selection** – where the input feature vector is multiplied element-wise with an input oracle mask, whose $i$th entry equals 1 iff the $i$th feature is relevant (e.g., on Syn1, features 1 and 2 are relevant, and on Syn4, features 1-6, and 11 are relevant), (2) **our (learned) feature selection mask** – where the input feature vector is multiplied element-wise with the mask $\mathbf{m}_t$, i.e., the entries of the mask $\mathbf{m}_s$ (see Section 2.3) are all fixed to 1, and (3) **no feature selection**.

From each dataset, we generated seven different instances that differ in their input size, $d \in [11, 50, 100, 150, 200, 250, 300]$. Where when the input dimension $d$ increases, the same logit function is used. Each instance contains 10k samples that were partitioned as described in Section D.1. We treated each instance as an independent dataset, and the grid search process that is described in Section D.1 was done for each one.

The FCN that we used has two dense hidden layers [64, 32] with a RELU activation. To keep things simple, we have not used drouput or any kind of regularization. The same training protocol was used for all three models. We used the same learning rate scheduler, early stopping protocol, loss function and optimizer as appear in Section D.2[9]. We use a batch size of 256, and an initial learning rate of 0.001. The only hyperparameter that was fine-tuned is the 'feature selection beta' in the case of 'FCN with feature selection' on the range $\{1.3, 1., 0.7, 0.4\}$. For the two other models, only a single configuration was tested in the grid search process.

---

[9]We noticed that in this scenario, a large learning rate or large batch size leads to a decline in the performance of the 'FCN with the feature selection'. While the simple FCN and the 'FCN with oracle mask' remains approximately the same.

