# OpenReview forum: "Net-DNF: Effective Deep Modeling of Tabular Data"
_ICLR.cc/2021/Conference — ICLR 2021 Poster_

### Official Review · AnonReviewer3 · 2020-10-25
**Novel neural architecture emulating decision trees/forest, but with gaps in empirical evaluation.**

**Rating:** 6
**Confidence:** 4

**Review:**

This paper proposes a neural architecture that emulates the characteristics of decision-tree variants, in the hope of mirroring their successes on tabular data. The architecture consists of three components: DNNF blocks, feature-selection masks, and spacial-localization weightings of DNNF blocks. I find these components and their coherent combination into the Net-DNF structure to be novel.

However, their empirical evaluations do not place the performance of Net-DNF in the context of existing work. It is good that the authors have compared Net-DNF against the obvious baselines of XGBoost and FCN, but they have neglected to demonstrate where Net-DNF stands in relation to other tabular-inspired approaches (mentioned in the related work). For example, why did the authors not compare against Popov et al. (2019; whose code is open-sourced) and Shavitt & Segal (2018)?

I feel that the authors have left out a line of work that uses a deep learning approach for tabular data. The reference below is a representative publication. (NB: The VAE approach can be evaluated on prediction tasks like Net-DNF, and it works on multi-modal data that Net-DNF purports to handle.) Could the authors comment on where Net-DNF stands relative to this line of work (and include it in their related work section)?

Nazabal, Alfredo, et al. "Handling incomplete heterogeneous data using vaes." Pattern Recognition (2020)

The exposition of the paper is lucid throughout.

QUESTIONS

* Page 1, para 2, "Moreover,...multi-modal data,...is problematic":
How does Net-DNF handle multi-modal data (e.g., "medical records and images")? Would it simply encode an image as a vector of bits and stack it in input x?

* Page 1, para 2, "GBDTs...hard to scale":
How scalable is Net-DNF? What is its size complexity?

* Page 6, Table 1 and 2.
"# formulas" corresponds to the number of DNNFs in an Net-DNF, but what does it correspond to in an FCN? How many parameters are in each system? Wouldn't the number of parameters provide a fairer comparison of the capacities of the systems?

* Page 8, Figure 2.
The authors claim that space localization will improve the green line's performance (feature selection) on Syn4-6. Why don't the authors provide empirical results to show this? Seems like a straightforward data point to add.

NITS

* Page 2, first line: inherent GBDTs -> inherent in GBDTs

* Page 6, In is evident -> it is evident

---

> ### Author Response · Authors · 2020-11-23
> **Response to AnonReviewer3**
>
> __*However, their empirical evaluations do not place the performance of Net-DNF in the context of existing work. It is good that the authors have compared Net-DNF against the obvious baselines of XGBoost and FCN, but they have neglected to demonstrate where Net-DNF stands in relation to other tabular-inspired approaches (mentioned in the related work). For example, why did the authors not compare against Popov et al. (2019; whose code is open-sourced) and Shavitt & Segal (2018)? Nazabal, Alfredo, et al. "Handling incomplete heterogeneous data using vaes." Pattern Recognition (2020)*__
>
> Early experiments with the method of Shavitt & Segal (using their own code) showed results that were substantially inferior and we decided to not pursue deeper exploration of their method. Popov’s approach is similar to FCNs according to their reported performance and we felt that it is reasonable to omit this additional comparison.
>
> __*I feel that the authors have left out a line of work that uses a deep learning approach for tabular data. The reference below is a representative publication. (NB: The VAE approach can be evaluated on prediction tasks like Net-DNF, and it works on multi-modal data that Net-DNF purports to handle.) Could the authors comment on where Net-DNF stands relative to this line of work (and include it in their related work section)?*__
>
> While the work cited by the reviewer is certainly relevant and we should have cited it, none of the experiments considered in the work are of a similar scale to ours. In particular, the categorical/tabular variables in each of the dataset considered is extremely small: just 1 for 4 datasets and 4,6 for the two additional datasets. Thus, the setting considered is mostly non-tabular in the categorical sense of the word. Moreover, the one-hot encoding they use to cope with tabular data, while reasonable for a small number of such variables, is less so when the domain has many categorical variables.
>
> __*Page 1, para 2, "Moreover,...multi-modal data,...is problematic": How does Net-DNF handle multi-modal data (e.g., "medical records and images")? Would it simply encode an image as a vector of bits and stack it in input x?*__
>
> We believe that our work is a step toward the ultimate goal of being able to fuse tabular data on which neural network perform poorly and other types of data (e.g. images) where NNs are the natural champions. One can envision a Net-DNF that parallels, for example a convolution network, where both representations are merged, perhaps fortified with mutual attention, and then classified/regressed using a couple of fully connected layers.  The crucial point is that such a setting allows for joined, end-to-end optimization of both modalities together.
>
> __*Page 1, para 2, "GBDTs...hard to scale": How scalable is Net-DNF? What is its size complexity?*__
>
> GBDTs and similar approach scale badly because they need to essentially store the entire data in memory. Net-DNF nets are similar to standard neural networks in that effective  optimization is carried via batches and methods like SGD. The effectiveness of this combination to learn strong models from huge datasets.
>
> from the paper: "Scaling up the gradient boosting models was addressed by several papers (Ye et al., 2009; Tyree et al., 2011; Fu et al., 2019; Vasiloudis et al., 2019). The most significant computational disadvantage of GBDTs is the need to store (almost) the entire dataset in memory."
>
> __*Page 6, Table 1 and 2. "# formulas" corresponds to the number of DNNFs in an Net-DNF, but what does it correspond to in an FCN? How many parameters are in each system? Wouldn't the number of parameters provide a fairer comparison of the capacities of the systems?*__
>
> Number of formulas is indeed meaningless in FCNs. See parameters count in response to AnonReviewer2
>
> __*Page 8, Figure 2. The authors claim that space localization will improve the green line's performance (feature selection) on Syn4-6. Why don't the authors provide empirical results to show this? Seems like a straightforward data point to add.*__
>
> We agree with the reviewer and, given the chance, will add this data point.

---

### Official Review · AnonReviewer2 · 2020-10-26

**Rating:** 7
**Confidence:** 2

**Review:**

The paper attempts to propose a neural network-based algorithm on tabular data, or non-distributed representation to address the unique challenges in tabular data, which are non-existing in the distributed data (images, language, etc). The primary motivation of this paper is the simulation of DNF, which simulates the Boolean formulas in decision making. The DNF form can better capture the non-distributional nature of tabular data and somehow simulates with the decision-tree algorithms in a more soft-threshold way. The general idea makes sense to me and sounds novel compared to the existing literature. However, I still have some doubts and questions about the paper:
1) In Sec 2.1, this soft approximation is a very critical part of the paper. Why do you decide to modify the constant from 1.0 to 1.5, you suggested in the Appendix that "the AND gate will be closer to 1", but the OR gate will be affected, right? Then OR(0,0,0,1) = tanh(0.5) = 0.4, which is pushed away from 1. I believe there are some trade-offs for this hyperparameter, how do you decide it should be 1.5 rather than other numbers, do you have supporting evidence for your decision?
2) In your feature selection part, the $w_t$ is a learnable parameter to control the sparsity of the selected feature. You propose to achieve this by using the elastic net regularization. Is the trainable mask $w_t$ shared among all the DNNF blocks or is it specific to a certain DNNF block?
3) In your experiment table 1, some results from FCN are shown to cause OOM. I kind of get the idea that DNNF uses a more compact parameterization, but not entirely sure how it quantitatively compare with FCN. What is the parameter complexity for DNNF? Could you give a more formal explanation about why it's more compact?
4) In table 3, some of the experimental results are better than XGBOOST and some are worse. I can understand that the current DNNF has not yet achieved the same performance as XGBOOST as XGBOOST has always been a go-to algorithm for tabular data. But could you explain why DNNF is much better than GAS Concentrations, is it because of some properties of this dataset? And do you have a more high-level suggestion as to "under what circumstances is DNNF likely to out-perform XGBOOST", what could be the reason for that? Is it because of the better generalization of the neural network?

Typo:
2.4 Spacial Localization -> Spatial Localization

---

> ### Author Response · Authors · 2020-11-23
> **Response to AnonReviewer2**
>
> __*In Sec 2.1, this soft approximation is a very critical part of the paper. Why do you decide to modify the constant from 1.0 to 1.5, you suggested in the Appendix that "the AND gate will be closer to 1", but the OR gate will be affected, right? Then OR(0,0,0,1) = tanh(0.5) = 0.4, which is pushed away from 1. I believe there are some trade-offs for this hyperparameter, how do you decide it should be 1.5 rather than other numbers, do you have supporting evidence for your decision?*__
>
> It is indeed likely that the value of the bias term has certain tradeoffs. Early on in our study we reasoned that it makes sense to emphasize the case of AND(1,1,...,1) and (arbitrarily) fixed it to 1.5 without further optimization (we realized that it can potentially hurt cases  like OR(1,0,0,...) and also AND(1,1,1,-1) = tanh(-5)). While we believe that this bias constant is not critical (because the first layer is trainable and can in principle compensate for any fixed bias term), in retrospect this was not a good choice due to symmetry arguments (and it justifiably raises doubts/questions like yours). At this stage it will be impossible for us  to repeat the entire empirical study with bias=1 in reasonable time, but we can add a quick examination that shows that results aren’t too sensitive on average to this constant.
>
> __*In your feature selection part, the  is a learnable parameter to control the sparsity of the selected feature. You propose to achieve this by using the elastic net regularization. Is the trainable mask  shared among all the DNNF blocks or is it specific to a certain DNNF block?*__
>
> $w_t$ is specific to each block (we will clarify this).
>
> __*In your experiment table 1, some results from FCN are shown to cause OOM. I kind of get the idea that DNNF uses a more compact parameterization, but not entirely sure how it quantitatively compare with FCN. What is the parameter complexity for DNNF? Could you give a more formal explanation about why it's more compact?*__
>
> The OOM was encountered when the DNF structure was replaced by a three hidden layer FCN. A fair comparison is obtained by defining the widths of the FCN layers in accordance with the widths in the Net-DNF with the given number of formulas (see Tables 1&2). To see the parameter complexity advantage of DNNF observe that in the pure DNNF block, only the first layer (which creates affine “literals” or features) is trainable. If d is the input dimension and we consider m literals, the difference is O(dm) for DNNF vs O(dm + m^2) for the corresponding FCN.
>
> __*In table 3, some of the experimental results are better than XGBOOST and some are worse. I can understand that the current DNNF has not yet achieved the same performance as XGBOOST as XGBOOST has always been a go-to algorithm for tabular data. But could you explain why DNNF is much better than GAS Concentrations, is it because of some properties of this dataset? And do you have a more high-level suggestion as to "under what circumstances is DNNF likely to out-perform XGBOOST", what could be the reason for that? Is it because of the better generalization of the neural network?*__
>
> This is an interesting open question for further research. While the inductive biases of XGBoost and Net-DNF both rely on a weighted sum of logical formulas, there are still substantial differences. For example, the formulas we use are over linear separators and XGBoost relies on decision stumps. Another major difference is the training process (SGD vs. decision tree learning and boosting). The family of generating distributions for tabular classification tasks is large and the labeling mechanisms can be diverse from very simple (e.g., labels defined by a very simple function of few input features) to very complex (e.g.,  parity-like). Thus, it is indeed likely that some datasets are better aligned with one representation or the other, but we believe that a-priori identification of what will work better is going to be quite elusive.

---

### Official Review · AnonReviewer4 · 2020-11-05
**Positioning w.r.t. TabNet is needed**

**Rating:** 6
**Confidence:** 3

**Review:**

The authors propose a end-to-end deep learning model called Net-DNF to handle tabular data. The architecture of Net-DNF has four layers: the first layer is a dense layer (learnable weights) with tanh activation eq(1). The second layer (DNNF) is formed by binary conjunctions over literals eq(2). The third layer is an embedding formed by n DNNF blocks  eq(3). the last layer is a linear transformation of the embedding with a sigmoid activation eq(4). The authors also propose a feature selection method based on a trainable binarized selection with a modified L1 and L2 regularization. In the experimental analysis, Net-DNF outperforms fully connected networks.

pros:

- A novel approach to handle tabular data using deep learning with an integrated feature selection
- The VC dimension bound gives theoretical motivation  for expressing decision trees as DNF formulas.

cons:

- Classical ML approaches are outperforming Net-DNF
- The experimental analyses are not convincing: The datasets are a bit limited and performance metrics are rather limited to log-loss which is difficult to interpret


remarks:

- It is unclear whether adding the Net-DNF layers, in the ablation study, gives improvement due to simply an increase of model capacity or the specific architecture of Net-DNF.
- In Exp 7, why there is OOM despite removing additional layer ?
- The authors did not discuss a similar recent work: TabNet (https://arxiv.org/abs/1908.07442)
- The authors need to include additional datasets similar to TabNet experiments
- in $R(m_t, m_s)$ I think there should not be a division by 2

---

> ### Author Response · Authors · 2020-11-23
> **Response to Reviewer4**
>
> __*Classical ML approaches are outperforming Net-DNF*__
>
> We do not claim to consistently beat classical gradient boosting trees methods on tabular data. Instead, we are making a real step towards a unified framework where you can be competitive with these on tabular data, while also offering the benefits of neural networks when the data is different or not amenable to such methods because of its size.
>
> __*The experimental analyses are not convincing: The datasets are a bit limited and performance metrics are rather limited to log-loss which is difficult to interpret*__
>
> We intentinally focused on large datasets where GBTs perform best, including two large historic Kaggle competitions whose sizes are 62k and 200K samples. As a result, the number of datasets is indeed limited because the computation we explored for the baselines, to ensure extra fairness. See details provided in the supplementary material (864 configurations for XGBoost, and 3300 configurations for FCNs).  We agree with the reviewer that log-loss isn’t natural to interpret. But,  it is the maximum likelihood function applied to test data and captures the extent to which the model fits the data. Importantly, this is the conventional/suggested metric on many of these tasks (check the *Kaggle* instructions for the competitions, e.g., the Otto competition https://www.kaggle.com/c/otto-group-product-classification-challenge/overview/evaluation).
>
> __*It is unclear whether adding the Net-DNF layers, in the ablation study, gives improvement due to simply an increase of model capacity or the specific architecture of Net-DNF.*__
>
> In terms of model capacity/parameters, some of the FCNs we considered were substantially larger (see response to Reviewer 2), but still were far behind in terms of error performance.
>
> __*In Exp 7, why there is OOM despite removing additional layer ?*__
>
> See response to Reviewer 2.
>
> __*The authors did not discuss a similar recent work: TabNet*__
>
> Thanks for the comment. We were not aware of this paper. We will add the following paragraph: TabNet’s paper also tackles the same problem, of providing a deep-learning based predictive model for tabular data comparative or superior to gradient boosting technique. The differences are in the technique used to reach this goal. TabNet uses an attention mechanism to induce sparsity/feature selection, while Net-DNF uses DNF formulas and elastic net regularization for feature selection.
>
> __*The authors need to include additional datasets similar to TabNet experiments*__
> We will add  TabNets experiments to the paper. We ran Tabnet on some of our datasets using the PyTorch implementation (https://github.com/dreamquark-ai/tabnet).  Tabnet results were slightly inferior to the results we obtained for XGBoost. For example, for the Gas Concentrations we got 4.89 log-loss (vs 2.22 on XGBoost).
>
> __*In $R(m_t, m_s)$ I think there should not be a division by 2*__
>
> This is how elastic net is derived and implemented. See, e.g. the sklearn implementation::https://scikit-learn.org/stable/modules/generated/sklearn.linear_model.ElasticNet.html.

---

### Decision · Program_Chairs · 2021-01-07
**Final Decision**

**Decision:**

Accept (Poster)

**Comment:**

The paper proposes an end-to-end architecture, Net-DNF,  for handling tabular data. This is a novel approach in the relatively under-explored domain of application of neural networks; the paper also presents justification of the design choices via ablation studies. The paper is clearly written, and empirical results are convincing.